# A Kinetic Transition Network Model Reveals the Diversity of Protein Dimer Formation Mechanisms

**DOI:** 10.3390/biom13121708

**Published:** 2023-11-26

**Authors:** Dániel Györffy, Péter Závodszky, András Szilágyi

**Affiliations:** 1Systems Biology of Reproduction Research Group, Institute of Enzymology, HUN-REN Research Centre for Natural Sciences, 1117 Budapest, Hungary; gyorffy.daniel@ttk.hu; 2Faculty of Information Technology and Bionics, Pázmány Péter Catholic University, 1083 Budapest, Hungary; 3Structural Biophysics Research Group, Institute of Enzymology, HUN-REN Research Centre for Natural Sciences, 1117 Budapest, Hungary; zavodszky.peter@ttk.hu

**Keywords:** protein folding, protein binding, kinetic network, homodimer, conformational selection, induced fit, Transition Path Theory

## Abstract

Protein homodimers have been classified as three-state or two-state dimers depending on whether a folded monomer forms before association, but the details of the folding–binding mechanisms are poorly understood. Kinetic transition networks of conformational states have provided insight into the folding mechanisms of monomeric proteins, but extending such a network to two protein chains is challenging as all the relative positions and orientations of the chains need to be included, greatly increasing the number of degrees of freedom. Here, we present a simplification of the problem by grouping all states of the two chains into two layers: a dissociated and an associated layer. We combined our two-layer approach with the Wako–Saito–Muñoz–Eaton method and used Transition Path Theory to investigate the dimer formation kinetics of eight homodimers. The analysis reveals a remarkable diversity of dimer formation mechanisms. Induced folding, conformational selection, and rigid docking are often simultaneously at work, and their contribution depends on the protein concentration. Pre-folded structural elements are always present at the moment of association, and asymmetric binding mechanisms are common. Our two-layer network approach can be combined with various methods that generate discrete states, yielding new insights into the kinetics and pathways of flexible binding processes.

## 1. Introduction

Proteins in living cells function as parts of molecular complexes. Understanding the formation mechanism of these complexes is one of the main challenges in molecular biophysics. Since Fischer proposed his “lock-and-key” model [1], molecular biologists have widely recognized the fact that molecular recognition is a dynamic process [2,3,4], and the function of protein complexes is determined not only by their native structure but also by their conformational dynamics. In some cases, understanding how a protein works requires the adequate sampling of a large ensemble of conformational substates, e.g., by molecular dynamics simulations, which is a formidable task for even a single protein chain due to the high dimensionality. If multiple chains are present, sampling all the possible combinations of the conformations of each chain, along with all their possible relative positions and orientations, quickly becomes computationally intractable.

In protein homodimers, the simplest form of protein complexes, three-state and two-state homodimers are distinguished based on whether the folding of the chains or their association occurs first. In the case of two-state homodimers, the system has only two experimentally observable stable states: the fully unfolded, dissociated state and the native dimer. Contrarily, three-state dimers have a third state, usually corresponding to the folded monomers.

The thermodynamics of protein dimerization can be investigated by analyzing the free-energy landscape, usually obtained by projecting the states of the system onto some predefined reaction coordinates, typically the proportion of native contacts (both intra- and intermolecular) or the root mean square distance (RMSD) from the native structure. Levy and co-workers [5,6,7] investigated two- and three-state protein complexes using a structure-based potential to enhance the sampling of the state space. This simplification made possible the derivation of free-energy landscapes as a function of certain folding and binding parameters as reaction coordinates. The free-energy landscape analysis could distinguish between two- and three-state dimers in agreement with the experimental findings, and showed topological models to be able to capture the main kinetic properties of folding and binding.

Like any other method, the free-energy landscape analysis method has some shortcomings. The main disadvantage of the method is that the projection of a large number of degrees of freedom onto a small number of coordinates can hide barriers between separated states, thus considering them as a single state. To overcome these problems, a fundamentally different method was recently proposed. Instead of a priori defining one or a few reaction coordinates, one can build a Markov State Model (MSM) from molecular dynamics (MD) simulation trajectories, which can then be analyzed statistically [8].

Protein folding and ligand binding were extensively investigated by MSMs constructed from MD simulation trajectories. MSMs can provide detailed insights into macromolecular processes not easily obtained by other methods. For example, Voelz et al. studied the folding of the first 39-residue segment of the fast folder protein NTL9 [9]. They constructed an MSM containing 100,000 microstates and then coarse-grained it into an MSM with 2000 macrostates as nodes. Investigation of the MSM revealed that the rate-limiting step in the formation of the β-sheet is the pairing of the β_1_ and β_2_ strands. In another study, they used an improved method to define 20 macrostates, and the resulting MSM revealed several parallel folding pathways [10]. They also investigated the folding of the D14A mutant of the 6–85 fragment of the 236-residue protein λ repressor by all-atom simulations [11] and created a high-resolution MSM with 30,000 states and a low-resolution one with 5000 states from the simulation data. The study found that the position of helix 5 in the native state differs from the one seen in the crystallographic structure and is mainly unstructured in most of the metastable states. The unfolded state was found to have a significant β-sheet content. A 10 ms timescale relaxation invisible in experiments was also revealed. Noé et al. found single-exponential kinetics for PinWW folding, with hairpin 2 forming before hairpin 1 formation [12].

The MSM method has proven useful for investigating protein–ligand interactions and protein complex formation. The relative roles of induced fit [2] and conformational selection [13], the two proposed mechanisms of coupled folding and binding [4], were assessed in the arginine binding of the LAO protein [14]. In the induced fit model, the ligand first binds to the unbound conformation of the protein, and this binding event induces the protein to transition to the bound state. In the conformational selection (or population shift) model, the intrinsic dynamics of the protein makes it constantly switch between a stable unbound conformation and a less stable bound conformation, and the ligand binds directly to the bound conformation, thereby stabilizing it. In the case of the LAO protein, an encounter complex was revealed, which can be formed by both induced fit and conformational selection. The native bound state was found to be formed from the encounter complex by induced fit. Another study examined the relative importance of induced fit and conformational selection for choline-binding by the ChoX protein [15]. Combining an MSM for the monomer protein, MD simulation for the ligand binding, and flux analysis, the authors found a strong dominance (90%) of conformational selection.

Contrarily, induced fit has proven to be the dominant mechanism in the folding of the p53 transactivation domain coupled with binding to MDM2 [16]. From an MSM containing 600 states, a *k*_on_ was calculated, which agreed well with the experimental value. Increasing the helix content of p53-TAD, a more significant contribution of conformational selection and a stronger binding were found.

The studies cited above show that MSMs can also model many-body systems. However, they require a non-physiologically high concentration of the reactants to obtain a sufficient number of binding events in MD simulations. Kelley et al. solved the problem of high concentration by extending their MSM obtained by MD simulations by analytically estimating association rate constants [17].

Zhou et al. demonstrated that MSM-based methods can be applied to protein dimer formation [16]. However, it is important to note that in that study, only one of the interacting partners had significant internal flexibility; the other partner was treated as a rigid body.

All the investigations cited above had a state space discretization step. Clustering conformations sampled during an MD simulation into microstates is, however, by no means trivial. Conformations need to be geometrically similar enough to be kinetically related, but the more similar the conformations in a particular macrostate, the larger the statistical error. Additionally, the discretized state space dynamics are not Markovian even when the original, continuous state space dynamics are Markovian [18]. We can avoid this by assuming that a projection of the full dynamics is observed on the discrete states, resulting in a Projected Markov Model, which may be estimated using a Hidden Markov Model [18]. An alternative way to avoid the discretization problem is using ensembles of discrete states generated by an ensemble-based method such as COREX [19] or the Wako–Saito–Muñoz–Eaton model [20,21,22,23]. If we can define a reversible move set for the states of the ensemble generated by a particular method, we can calculate the probabilities of transitions by the Metropolis–Hastings criterion [24,25].

Building a Markov State Model based on an intrinsically discrete state space requires exploring all possible states. For a system consisting of two protein chains, the number of possible states is huge, not primarily because of the large number of possible conformational states but because of the large number of possible relative positions and orientations. Thus, to make the system tractable, we need to reduce the state space by eliminating the large number of states only differing in the relative positions and orientations of the chains.

To reduce the size of the state space, we propose a two-layer kinetic network model that makes it possible to study the homodimer formation of proteins using an MSM method. In the two-layer model, two states belong to every pair of conformational states: one where the two chains are associated and another where they are dissociated.

To demonstrate the applicability of the model, we applied our two-layer kinetic network model to a Wako–Saito–Muñoz–Eaton (WSME) representation [21,22,23] of several three- and two-state protein complexes. Using Transition Path Theory [26,27], we were able to characterize in a detailed manner the kinetics of the dimer formation of some homodimers that have not been previously investigated kinetically.

## 2. Methods

### 2.1. Studied Structures

Eight homodimers, the folding and binding of which were studied before by Levy et al. [7], were selected, including four two-state and four three-state homodimers (see Table 1). For each homodimer (where it was available), the first biological assembly was downloaded from the PDB database (https://rcsb.org, accessed on 29 June 2021). The hydrogen and heteroatoms were removed from the structure, and the missing residues were filled by dummy residues. A fixed number, *k*, of consecutive residues were combined into folding segments. If the number of residues in the whole sequence was not divisible by *k*, the C-terminus was extended by dummy residues to make the chain length a multiple of *k*. Dummy residues were considered as residues in a coil conformation in the native dimer and had a zero folding entropy term in the Wako–Saito–Muñoz–Eaton Model.

### 2.2. Wako–Saito–Muñoz–Eaton Model

The WSME model [21,22,23] is an Ising-like model for generating denatured ensembles of proteins. In this model, each conformational state of a protein is characterized by the folding state of the residues. A particular amino acid can have two states: folded or unfolded, and the conformational state of the whole protein can be characterized by the states of single residues (or sections of the chain consisting of multiple residues). The free energy of the conformational state *v* is calculated by
$$F\left(v\right)=E-TS=\sum_{\left(i,j\right)\in N}^{}\epsilon_{ij}\prod_{k=i}^{j}v_{k}-T\sum_{i=1}^{n}s_{i}\left(1-v_{i}\right)$$
where *N*, ϵ*_ij_* and *s_i_* are the set of native contacts, the energy of the native contact between residue (segment) *i* and *j*, and the entropy gain occurring when a residue unfolds, respectively, and *v_i_* = 1 if the *i*th residue is in the folded state and *v_i_* = 0 otherwise.

We extended the WSME model to two chains, considering not only the intramolecular but also the intermolecular interactions. In this model, intermolecular interactions are treated the same way as intramolecular interactions. The model is constructed from the crystal structure of the dimer. Two residues are considered to be in contact if the distance of at least one pair of their atoms (one atom from each chain) is < 4 Å. For intramolecular contacts, a contact between the *i* and *j* residues is assigned a non-zero energy term if at least two residues separate *i* and *j* along the chain, and all the residues between *i* and *j* (*i* and *j* included) are in the folded state (as in the original WSME model), ensuring the coupling between the folding state of neighboring amino acids. For intermolecular contacts, a contact between residues *i* and *j* is assigned a non-zero energy term if both residues are in the folded state. Each dissociated state was assigned a positive dissociation entropy term of 87 J/K/mol, calculated according to Tamura and Privalov [28] assuming a 100 μM solution of monomers. Although there are significantly different estimates in the literature for the entropy cost of association, this does not affect our results as we have investigated the processes at a wide range of dissociation entropy values.

The total free energy of the two-chain system is
$$F\left(u,v,\delta \right)=F\left(u\right)+F\left(v\right)+\delta \cdot \sum_{\left(i,j\right)\in M}^{}\epsilon_{ij}\cdot u_{i}\cdot v_{j}+{\left(1-\delta \right)\cdot S}_{\mathrm{diss}}$$
where *u* and *v* are the conformational states of the two chains, respectively, *M* is the set of native contacts between the chains, respectively, *S*_diss_ is the dissociation entropy term, and δ = 1 if the two chains are associated and δ = 0 if they are dissociated.

Here, we applied the single sequence approximation for the individual chains, i.e., where only one continuous stretch of residues can be folded. Even though this is less accurate than the double or triple sequence approximations [29], using the double sequence approximation would have increased the number of states from 8978 to 631,688 (with 11 segments) or from 12,428 to 1,260,872 (with 12 segments). As the size of the transition matrix is the square of the number of states, this would have been computationally prohibitive. The further details of the model were set according to ref. [21], but some parameter values were slightly changed. The temperature was 310 K, and the value of ε, which is the elementary contact energy between residues, was calculated for each dimer so that the probability of the native dimer is 0.5. (The obtained values for ε in Joules per mole were: 1arr: 1840.1, 1bsr: 1253.1, 1cop: 868.0, 1cta: 1367.7, 1fia: 1388, 1lfb: 1356.4, 1lmb: 1326.6, 2oz9: 1470.4). The folding entropy term was set to zero for residues being assigned a blank (indicating no secondary structure) by DSSP [30]. For performance reasons, we used short segments rather than single residues as basic folding units. We choose the length of the basic folding units so that the number of folding units in the particular chains is approximately the same (11 or 12 here).

### 2.3. Transition Network

For studying the dynamics of our systems, we defined a transition network, a weighted directed multigraph where the nodes are the microstates of the system and the edges correspond to elements of a predefined move set. Each microstate of a protein homodimer is fully determined by the conformational states of monomers and their relative positions. In the WSME framework, when using the single sequence approximation, the conformational states of monomers are characterized by the length and position of the folded stretch. Transitions between conformational states were defined as elongation or shortening of the folded stretch by one folding unit. A single step in the state space is either a conformational transition of one of the two chains or an association/dissociation step. The weight of a particular edge was defined as the conditional probability of a transition through that edge. The transition probabilities were calculated analogously to the Metropolis–Hastings criterion [25] used in Markov Chain Monte Carlo simulations, which gives the probability *p_ij_* of a transition from state *i* to state *j* as
$$p_{ij}=min\left(1,\frac{p_{ji}^{\left(ap\right)}}{p_{ij}^{\left(ap\right)}}e^{\left(E_{i}-E_{j}\right)/RT}\right)p_{ij}^{\left(ap\right)}$$
where *p_ij_*^(*ap*)^ is the *a priori* probability of a transition to the state *j* being in the state *i*. Since the move set used here is fully reversible, *p_ij_*^(*ap*)^ can be replaced by 1/*n_i_* where *n_i_* is the outdegree of state *i* [31], so that the expression for the transition probability becomes
$$p_{ij}=min\left(1,\frac{n_{i}}{n_{j}}e^{\left(E_{i}-E_{j}\right)/RT}\right)/n_{i}.$$

### 2.4. Transition Path Theory

The rate constants of elementary reactions are commonly derived theoretically using the well-known transition-state theory (TST). In TST, the transition state is a saddle point on the potential energy surface, which has to be crossed by the system for a successful reaction to occur. However, in the case of complex systems, where entropic effects are important, the transition state may not correspond to a saddle point. More importantly, not all barrier-crossing events lead to a reaction. Transition Path Theory (TPT) is a theoretical framework that enables the derivation of reaction rates and dominant reaction paths given the fluxes between particular adjacent states in the state space *(S)*, whether it is continuous [27] or discrete [26]. TPT considers only transition paths, i.e., trajectories corresponding to successful transitions from the reactant set *(R)* to the product set *(P)* of microstates, and statistically analyzes the transition path ensemble.

The key concepts of TPT are the forward and backward committor functions. The forward committor function specifies the probability that the system, starting from a particular microstate, will reach the product state before reaching the reactant state. The backward committor function specifies the probability that the system, arriving at a particular microstate, came from the reactant state rather than the product state.

The reaction flux *f_ij_* from microstate *i* to *j* depends on the equilibrium probability of *i*, the committor function of *i*, and the transition probability from *i* to *j*.

## 3. Results

### 3.1. Two-Layer Network Model

Our purpose was to construct a model that allowed us to perform exact calculations describing the kinetics of protein homodimer formation. The transition network derived directly from the WSME model would be too large to be computationally tractable, mainly because of the large number of states that differ only by the relative positions and orientations of two chains. To handle a system with such a large number of microstates, one has to apply some simplification to reduce the size of the state space.

To reach a significant reduction in the number of microstates, we introduce a two-layer network model (Figure 1) obtained by (i) merging the dissociated microstates of each conformation pair into a single state and (ii) merging the associated microstates of each conformation pair into a single state, corresponding to the minimum-energy associated state of the particular conformation pair, where “conformation pair” refers to the combination (*i, j*) when one chain is in conformation *i* and the other in conformation *j*. The dissociated states of a particular conformation pair can be merged as these states only differ in the relative position and orientation of the chains and, thus, are energetically equivalent (excluding long-range interactions between distant chains). The merging of the associated states of a particular conformation pair is based on the assumption that only the minimum-energy state is significantly populated; this corresponds to the assumption that the partial energy landscape of binding between the particular conformations is funneled [32]. Thus, two states are assigned each conformation pair: an associated and a dissociated one, and the associated/dissociated states form the two layers of the network, as illustrated in Figure 1.

For this study, the free energies of the states were calculated according to the WSME model as described in the Methods section.

### 3.2. Mechanisms of Dimer Formation

For protein homodimer formation, we distinguish three mechanisms by the folding state of the monomers at the moment of association: (1) If both chains are folded when they bind (folding before binding), we call the underlying mechanism *rigid docking*. (2) When exactly one of the two chains is in the native conformation when binding, we call it *conformational selection*. (3) Finally, we call it *induced folding* when neither of the chains is folded when they bind (binding before folding). These mechanisms are illustrated in Figure 2.

Usually, two-state homodimers are thought to form by induced folding, while three-state homodimers by rigid docking, while conformational selection may fall in either category. To obtain a more detailed insight, we investigated eight homodimers, four classified as two-state and four as three-state homodimers [7], and performed transition path calculations using the Transition Path Theory proposed by E and Vanden-Eijnden [26,27]. We defined the reactant set as the union of all dissociated states while the product set as the states where all native contacts were present (this is not necessarily a single state as there may be folding units without contacts). Thus, the transition paths from the reactant to the product set reflect the transitions where the system reaches the native state after association without dissociating again. Each edge going from a dissociated to the corresponding associated state was assigned to one of the three mechanisms, and the sum of the reactive fluxes over edges belonging to each particular mechanism was calculated. By dividing the reactive fluxes calculated for each mechanism by the total reactive flux, we can calculate the relative contribution of each mechanism to the dimer formation as follows:$$p_{\mathrm{induced}\;\mathrm{folding}}=\frac{1}{f_{\mathrm{total}}}\sum_{i\in U+U,j\in UU}f_{ij}$$
$$p_{\mathrm{conformational}\;\mathrm{selection}}=\frac{1}{f_{\mathrm{total}}}\sum_{i\in U+F,j\in \mathrm{UF}}f_{ij}$$
$$p_{\mathrm{rigid}\;\mathrm{docking}}=\frac{1}{f_{\mathrm{total}}}\sum_{i\in F+F,j\in FF}f_{ij}$$
where
$$f_{\mathrm{total}}=\sum_{i\in C^{\mathrm{diss}},j\in C^{\mathrm{ass}}}f_{ij}$$
is the total reactive flux from the dissociated to the associated states, $C^{\mathrm{diss}}$ and $C^{\mathrm{ass}}$ are the set of dissociated and associated states, respectively, and $f_{ij}$ is the reactive flux, calculated by the Transition Path Theory, flowing from the state *i* to state *j.* U and F denote the unfolded and folded states of each chain, and U+U, UU, etc., denote the sets of various combinations of folded and unfolded and dissociated and associated states, as indicated in Figure 2. Based on these relative contributions, we could determine which mechanism is dominant (if any).

Table 1 shows the relative contributions of the three mechanisms for the eight homodimers. Interestingly, one mechanism strongly dominates for some dimers, but two or all three mechanisms have comparable contributions for others. Induced folding is the dominant mechanism for three of the four two-state dimers, and rigid docking is the dominant mechanism for two of the four three-state dimers. For the two-state troponin C site III (1cta), conformational selection dominates but induced folding still contributes more than rigid docking. For the three-state 𝜆-Cro repressor (1cop), conformational selection dominates as well but rigid docking still has a much higher contribution than induced folding.

The bovine seminal ribonuclease (BS-RNase, 1bsr) is a special case as it has been classified as a three-state dimer [7], but we find that induced folding is dominant. This enzyme is known to be able to form both a domain-swapped dimer and a non-swapped dimer [33], which should have different folding kinetics. The two forms are in equilibrium in solution, with a swapped/non-swapped ratio of about 7:3 for the wild-type enzyme and about 1:1 for the PALQ double mutant [34]. While the non-swapped dimer can form by rigid docking, the swapped dimer cannot as the monomer must open up and cross over to the other subunit. The 1bsr structure represents the domain-swapped form, and we used the separated chains of this form as the folded monomer structure in our method; therefore, our results describe dimer formation assuming that the open monomer form is the native monomeric state. We also performed our transition path analysis on a non-domain-swapped form of BS-RNase (PDB ID: 3bcm), and, in accordance with our expectations, we found that rigid docking was the mechanism with the highest flux by a large margin (results not shown). In reality, the dimer formation mechanism is a mixture of those for the two forms, and a full description of its complexity is beyond the scope of our study.biomolecules-13-01708-t001_Table 1Table 1Relative contribution of the three mechanisms (induced folding, conformational selection, and rigid docking) to the dimer formation of 8 homodimers.Protein Name/PDB ID (States)Induced FoldingConformational SelectionRigid DockingArc repressor/1arr (2)**0.9897**1.029 × 10^−2^2.100 × 10^−5^Troponin C site III/1cta (2)0.2489**0.6102**0.141Factor for inversion stimulation/1fia (2)**0.9311**6.889 × 10^−2^2.398 × 10^−7^Trp repressor/2oz9 (2)**0.9997**3.313 × 10^−4^9.611 × 10^−11^BS-RNase/1bsr (3)**0.9986**1.424 × 10^−3^1.429 × 10^−7^λ Cro repressor/1cop (3)1.376 × 10^−3^**0.6994**0.2993LFB1 transcription factor/1lfb (3)4.096 × 10^−9^1.341 × 10^−4^**0.9999**λ repressor/1lmb (3)4.080 × 10^−4^7.295 × 10^−2^**0.9266**The dominant mechanism is indicated in bold.

### 3.3. Folding Degree of Binding Chains

In our two-chain WSME model, it is possible to calculate the folding degree, *Q*, of each chain as the number of folding units that are folded in the given conformation. We can then obtain the reactive fluxes flowing from the dissociated to the associated state as a function of the folding degrees of the chains by
$$F\left(Q_{A},Q_{B}\right)=\frac{1}{f_{\mathrm{total}}}\sum_{i\in C_{Q_{A},Q_{B}}^{\mathrm{diss}},\;j\in C_{Q_{A},Q_{B}}^{\mathrm{ass}}}f_{ij}$$
where $C_{Q_{A},Q_{B}}^{\mathrm{diss}}$ and $C_{Q_{A},Q_{B}}^{\mathrm{ass}}$ are the set of dissociated and associated microstates, respectively, for which the folding degree of the chain A is $Q_{A}$, and the folding degree of the chain B is $Q_{B}$, and $f_{ij}$ is the reactive flux flowing from state *i* to state *j.*

The heatmaps in Figure 3 visualize these fluxes. The heatmaps represent the distribution of binding events over the folding degrees of the two chains; the locations of the red areas on a heatmap show us the folding degrees of the chains when most binding events occur. In the heatmap, rigid docking appears in the bottom right corner, conformational selection at the bottom and the right edges of the map, and induced folding appears away from these areas. Thus, the heatmap provides us with a finer-grained picture of the dimer formation process than the simple classification into three mechanisms.

A considerable diversity is observed even within each category. As expected, the chains are more folded at the moment of association in three-state dimers than in two-state dimers, although in most cases, the chains are not fully folded. The slight asymmetry of some of the heatmaps (e.g., 1fia) results from the structural asymmetry of the dimers. Among the two-state dimers (Figure 3A), factor for inversion stimulation (1fia) and Trp repressor (2oz9) can only associate while partially folded, i.e., the fully folded monomers are virtually unable to bind, or occur with very low probabilities in the dissociated state. The heatmap of Arc repressor (1arr) shows the binding of partially folded monomers, which agrees with the findings that Arc repressor forms a molten globule in the monomer state [35,36]. The monomers of Troponin C site III (1cta) can bind easily in fully folded states, but also in many partially folded states. Among three-state dimers (Figure 3B), LFB1 transcription factor (1lfb) demonstrates a clear-cut rigid docking behavior, while the monomers of λ Cro repressor (1cop) can also bind when only one chain is highly folded (conformational selection). The binding of the chains of λ repressor (1lmb) can occur between two (almost) fully folded monomers or in states where one is almost fully folded while the other is in a very unfolded state. In their free-energy calculations, Levy et al. [7] found stable states of the λ repressor where one or both chains are in their folded conformation. These stable states can serve as the origin of association, as seen in our calculations. Finally, the heatmap of BS-RNase (1bsr) reveals the complexity of the formation mechanism of the domain-swapped dimer as there are three distinct hotspots in the map with one or both chains partially folded.

Notably, all heatmaps indicate the existence of asymmetric binding mechanisms as we usually see significant flux outside the main diagonal of the heatmap, indicating scenarios where the two chains are in very different states at association. This asymmetry is usually stronger for the three-state dimers.

### 3.4. Pre-Folded Segments

It is possible in our framework to determine which pre-folded structural elements are present in the unfolded proteins at the moment of association. To determine how folded the particular segments of the protein chains before binding are, we calculated for each segment *k* the probability that it is in the folded conformation at the moment of association by
$$F_{k}=\frac{1}{f_{\mathrm{total}}}\sum_{i\in C_{k}^{\mathrm{diss}},j\in C_{k}^{\mathrm{ass}}}f_{ij}$$
where $C_{k}^{\mathrm{diss}}$ and $C_{k}^{\mathrm{ass}}$ are the set of dissociated and associated states, respectively, where the *k*th segment is folded, and *f_ij_* is the reactive flux from state *i* to state *j*.

Figure 4 shows the foldedness probability maps for the investigated complexes. Looking at two-state dimers (Figure 4A), all appear to contain some fully folded segments at the moment of association, mostly α-helical regions, which fold the fastest in the monomers. Three-state dimers (Figure 4B) show mostly folded chains, except bovine seminal ribonuclease (1bsr), which we have already discussed.

### 3.5. Relative Weights of Mechanisms vs. Concentration

As we expect, the relative importance of the dimer formation mechanisms varies with protein concentration. In our model, we can mimic concentration change by altering the dissociation entropy term. Lower dissociation entropy is equivalent to higher concentration. Figure 5 shows the variation of the relative weight of each mechanism with protein concentration for the studied dimers. For each dimer, as the concentration decreases, the relative weight of rigid docking increases while that of induced folding decreases. In several cases, conformational selection initially increases and then decreases, i.e., it has a maximum at a particular concentration.

The effect of varying the concentration can also be observed at the level of residues, as shown in Figure 6 for the Arc repressor (1arr) as an example. It can be seen that the lower the concentration (or higher the dissociation entropy term), the higher the probability of the residues being folded at the moment of association, as the chains have more time to fold before they bind.

## 4. Discussion

Modeling folding and binding processes in protein dimer formation are challenging for several reasons, one being the large number of dimensions describing the relative positions and orientations of the chains involved in the recognition process. We have introduced a new kinetic transition network-based model for investigating protein dimer formation mechanisms. The model relies on two assumptions: (i) the dissociated states of a conformation pair are energetically equivalent (the energy does not depend on the relative position and orientation of the chains) and, therefore, they can be merged for each conformation pair, and (ii) among the associated states, only the one with the lowest energy is populated significantly; therefore, the states with higher energies can be omitted. With these assumptions, we constructed a transition network containing two nodes for each conformation pair of the two chains: an associated and a dissociated one. Thus, the properties of all the associated and dissociated states for the particular conformation pair are condensed into a single associated and a single dissociated state.

This reduction of the conformational space has allowed us to build a Markov model for the dynamics of a two-chain system and study its properties in an exact manner. It should be noted that although we used the structure-based WSME model (extended to two chains) to investigate dimer formation mechanisms, the two-layer transition network model is not limited to structure-based models and may be applicable to other types of models as well.

The investigation of our exact model describing the folding and binding dynamics of the system in terms of a kinetic transition network allows us to gain insight into the detailed mechanism of protein dimer formation. Using this model, we can not only determine whether the folding or the binding occurs first, but also the probabilities of all possible pathways and the probability of dimer formation through particular structures.

From our investigations of eight homodimers, the following main conclusions can be made:

(1) We have found a remarkable diversity of dimer formation mechanisms, as illustrated by the heatmaps in Figure 3. This indicates, in agreement with earlier findings [5,6,7] that the conventional classification of dimers into just two categories (two-state and three-state) is inadequate to describe the multitude of binding mechanisms. Even the three categories we defined (induced folding, conformational selection, and rigid docking) only provide a very rough picture of the actual binding mechanism. It is also notable that most dimer formation pathways involve asymmetric binding mechanisms (where the two chains are in very different conformations when binding).

(2) When classifying the dimer formation pathways into the three categories illustrated in Figure 1 and quantifying their contribution to the dimer formation process for each protein, we found that all three types of pathways are simultaneously at work. While one pathway often dominates over the others, there are cases when the three types of pathways have similar weights. Although some two-state dimers use mostly induced folding and some three-state dimers mostly use rigid docking, there are exceptions to this rule, and conformational selection can dominate in both categories.

(3) The relative probabilities of the three types of pathways are concentration dependent. The dominant mechanism usually varies from rigid docking at low concentrations to induced folding at high concentrations, with conformational selection often dominating at medium concentrations. The change of the dominant mechanism with varying concentration suggests that with the conventional classification of dimers into two- and three-state one may depend on the experimental conditions. Our finding that the binding mechanism is concentration-dependent is in accordance with the findings of earlier studies of protein–ligand binding [37,38,39,40,41,42] and the binding of intrinsically disordered proteins to their targets [43,44], although in these cases, only one of the binding partners is flexible. Of note, Hammes et al. [37] pointed out that the dominant binding mechanism can only be determined using flux methods (like our study) rather than from rate constants.

(4) Pre-folded structural elements are virtually always present at the moment of association, even for two-state dimers with induced folding as their dominant mechanism. The pre-folded elements tend to be α-helical segments. This finding supports the role of preformed structural elements in molecular recognition [45].

Levy et al. [7] investigated the same homodimers as our study, using simulations performed with an off-lattice Go model. Their main finding, namely that the overall binding mechanism and the nature of the binding transition state ensemble can be understood from the network of interactions that stabilize the native fold, is in agreement with our results. The free-energy landscapes they constructed made possible the thermodynamic characterization of the systems, including the transition states, the presence of intermediate states and the temporal order of folding and binding events. Our flux-based method is complementary to the free-energy landscape approach, as it provides a description of processes in terms of kinetics and transition pathways, thus revealing the simultaneous presence of alternative folding–binding orders for a particular dimer and the microstate level pathways and their statistical weights during dimer formation. Remarkably, our heatmaps representing the fluxes from the dissociated to the associated states show significant agreement with the three-dimensional free-energy landscapes calculated by Levy et al. [7] (Figure 2 and Figure 5 in their paper). Regions with high flux on our heatmaps roughly correspond to regions with low free energy on the landscapes. For instance, for Trp repressor (2oz9), our heatmap shows high flux in the region where both chains are highly unfolded, whereas in the free-energy landscapes, states with high folding degrees have low free energy only where the binding reaction coordinate is also high. Conversely, for LFB1 transcription factor (1lfb), the bound state can only be reached with almost fully folded chains, according to both our heatmap and the free-energy landscape. For Arc repressor (1arr) and troponin C site III (1cta), a large range of folding degrees contributes high reactive fluxes to the dimer formation, which is reflected in Levy et al.’s landscapes as the broad “tunnel” of low free-energy regions from the fully unfolded and unbound state to the native dimer. A significant asymmetry of folding states of the chains during binding can also be observed in the landscapes of factor for inversion stimulation (1fia) and λ repressor (1lmb), and to a lesser extent in the landscape of λ Cro repressor (1cop), similarly to our heatmaps.

The limitations of our study include the fact that the two-layer transition network requires a discretization of states, neglects potential long-range interactions between dissociated states, and only considers the lowest-energy associated state for each conformation pair. Furthermore, the WSME method is structure-based and ignores non-native interactions. Also, for computational efficiency, we had to use the single sequence approximation and segments longer than one residue as a folding unit.

As this is a computational study, it is hypothesis-generating by nature, and needs experimental verification. A wide range of experimental methods are available to study and discriminate protein binding mechanisms, including measurements of kinetics while varying the concentration of reactants, NMR, single-molecule fluorescence resonance energy transfer (smFRET) techniques, and measurement of Φ values (the relative destabilization by mutants) [43,46]. However, these techniques have mostly been applied to single chains or binding events where only one of the partners is flexible. The experimental characterization of (potentially asymmetric) folding/binding events of homodimers constitutes a special challenge, and simulations, combined with experiments, can provide unique insight into the details of molecular processes [41].

Although we applied our two-layer network approach in combination with an ensemble-based model (WSME) [20], we could similarly use it with other ensemble-based methods such as COREX [19]. It could also be combined with other representations of protein structure and dynamics, especially simple exact protein models such as the two-dimensional HP lattice model [47]. Generally, our two-layer model can be combined with sampling methods where the conformational space of each chain is discretized, and the minimum energy of inter-chain interactions for a conformation pair can be calculated (or at least estimated). For example, one can sample the conformational space of the monomers by MD simulations, and the conformational states can be discretized, e.g., by clustering. Then, the minimum inter-chain interaction energies can be estimated by rigid body docking for every conformation pair. From these data, a two-layer transition network can be built and analyzed, yielding new insights into the kinetics and pathways of various flexible binding mechanisms. Exploring these potential applications of the two-layer transition network approach will be the subject of future studies.

## Figures and Tables

**Figure 1 biomolecules-13-01708-f001:**
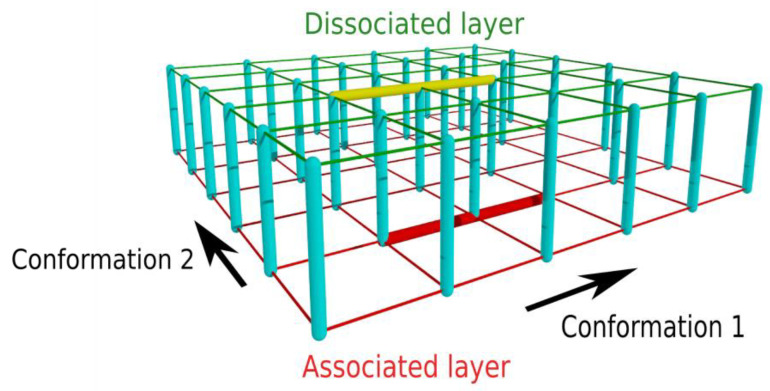
Illustration of the two-layer kinetic network model. The associated (dissociated) states form the bottom (top) layer. Transitions within each layer (thick yellow and red lines) correspond to conformational changes of the chains, while edges connecting the two layers (thick cyan lines) represent association/dissociation of the chains while their conformation is unchanged.

**Figure 2 biomolecules-13-01708-f002:**
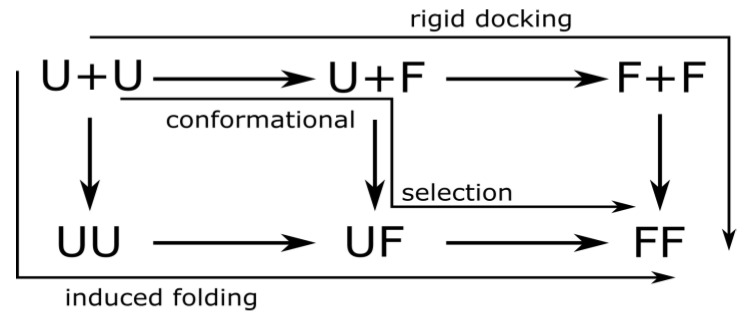
Illustration of the three mechanisms of dimer formation. U and F denote an unfolded and a folded chain, respectively. Horizontal arrows represent folding of a chain while vertical arrows represent association of the chains. The three mechanisms correspond to three distinct paths of getting from U+U (two unfolded dissociated chains) to FF (the folded dimer).

**Figure 3 biomolecules-13-01708-f003:**
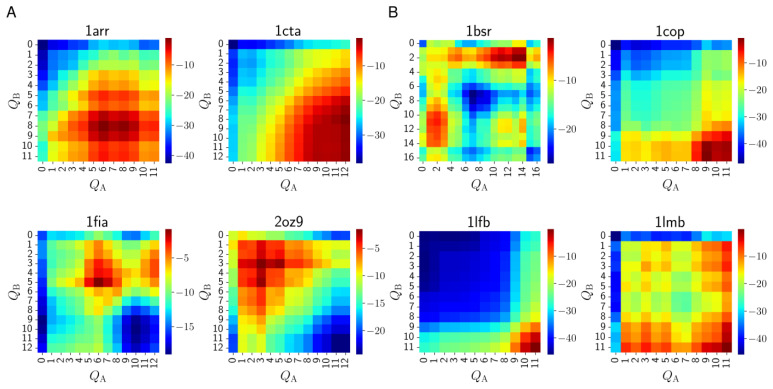
Heatmaps representing the fluxes of association as a function of the folding degrees (*Q*_A_ and *Q*_B_) of the two chains. The fluxes are normalized to 1, and the color scale corresponds to the logarithm of the relative flux. The folding degree is defined as the number of segments that are folded. (**A**) two-state dimers (**B**) three-state dimers.

**Figure 4 biomolecules-13-01708-f004:**
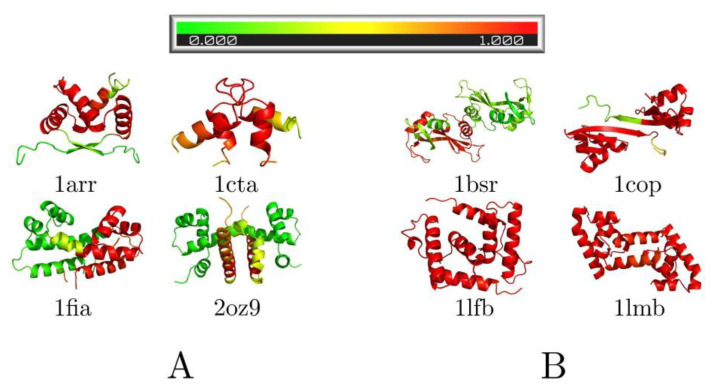
The foldedness probability of particular residues at the moment of association for (**A**) the two-state and (**B**) the three-state dimers. Two-state dimers are rather unfolded when the two chains associate, while three-state dimers are essentially fully folded, except for bovine seminal ribonuclease (PDB: 1bsr, see explanation in the main text).

**Figure 5 biomolecules-13-01708-f005:**
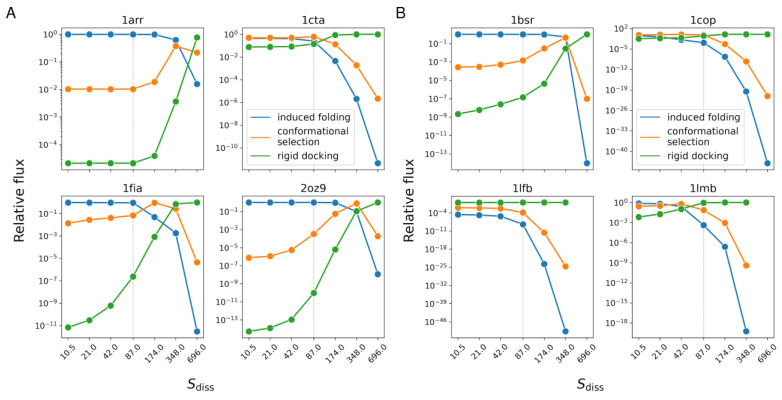
Probabilities of the three dimer formation mechanisms (induced folding, conformational selection, and rigid docking) as a function of the dissociation entropy term for the two-state (**A**) and the three-state (**B**) dimers. While at low values of dissociation entropy (high concentration), the induced folding is dominant, at high entropy values (low concentration), rigid docking is the primary mechanism. In some cases, conformational selection dominates at medium concentrations. The dashed lines represent $S_{diss}=$ 87 J/K/mol, the dissociation entropy value calculated theoretically [28] and used in other calculations.

**Figure 6 biomolecules-13-01708-f006:**
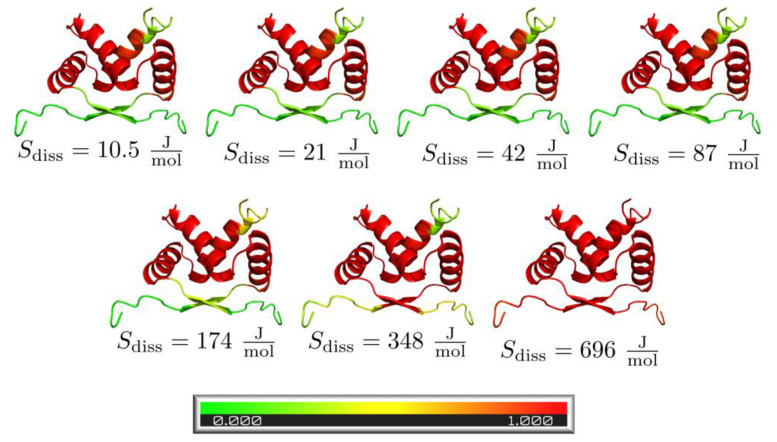
The effect of protein concentration on the foldedness probability of Arc repressor (1arr) residues at the moment of association. As concentration decreases (dissociation entropy increases), the foldedness probabilities increase.

## Data Availability

The data presented in this study are openly available at https://github.com/gydlacf/two_layer_paper (accessed on 22 November 2023).

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
