# Peer review of "A Kinetic Transition Network Model Reveals the Diversity of Protein Dimer Formation Mechanisms"

_biomolecules, 2023, doi:10.3390/biom13121708_

Round 1
Reviewer 1 Report
Comments and Suggestions for Authors
Review of biomolecules-2732022 paper.
The paper presented for my review is a nice example of a kinetic simulation of protein folding/binding with the help of Wako-Saito-Munoz-Eaton model.
Details of simulations and approach are presented well. I do not see a contradiction in obtained results, but their in-depth discussion and comparison with the results of other authors would increase the value of the manuscript. After a revision, the paper is suitable for publication.
Below please find my comments.
Sincerely,
Reviewer
---------------------------------
A technical comment: all the figures should be made roughly twice smaller.
l. 67-68 "For example, Voelz et al.
studied the folding of the first 39-residue segment of the fast folder protein NTL9 [9]."
Voelz et al are not the only authors, who has studied the kinetics of folding of the 39-residue segment of NTL9.
For instance, 2 years before them, MD simulations of a coarse-grained model have shown non-two-state kinetics of the same system (https://doi.org/10.1016/j.jmb.2008.09.023).
l.81-83 "The relative roles of induced fit [2] and conformational selec-
tion [12], the two proposed mechanisms of coupled folding and binding [4], were assessed
in the arginine binding of the LAO protein [13]."
It would be very useful for the reader, if the authors explain the meanings of "induced fit" and "conformational selec-
tion", and not just to cite. Could be as a footnote, or two sentences in the text.
l. 176 "Here, we applied the single sequence approximation..."
Please provide the justification/reasoning for the use of the single-sequence approximation used.
For instance, for a Zimm-Bragg model it fails to describe the system, when the length exceeds the scale of correlations (rigidity), which is of the order of several residues for polypeptides (see Fig.2 of https://doi.org/10.3390/polym13121985).
l. 282 "Table 1 shows the relative contributions of the three mechanisms..." and also in captions of Table 1:
Please define in the text, and mention in captions, how exactly are the "relative contributions" calculated.
l. 312 "In our two-chain WSME model, it is possible to calculate the foldedness of each chain..."
May I suggest to use "the degree of folding" or "folding degree" instead? Never seen "foldedness" in other papers. Also, since the WSME model is a native-centric approach, it is possible to use a standard, "native contact number" or "degree" expression, and not to introduce a new terminology for a well-known quantity.
General comments/recommendations to the authors:
1. It would be nice to provide wider comparison with the results of other authors.
2. The conclusion (3), about the concentration-dependence of the mechanism, I believe, is very important and deserves to be discussed and compared with the results of others.
Reviewer 2 Report
Comments and Suggestions for Authors
Gyoffry et al has simplified application of a Markov model to dimers, by assuming the a conformational state of the two molecules can be represented by one completely dissociated state, and one associated state of minimal energy (no partially or not minimized energy associations). This makes the computation doable. They treat 8 homodimers, and describe how the computation predicts they would form: rigid docking, induced fold, or conformational selection, giving weights to the three mechanisms. They go further to sub-classify the folding mechanisms using heat maps, and show the dependence of folding mechanism upon concentration.
The presentation is clear (except for the heat maps, see below) and methods seem appropriate. I am not a computational chemist, but from my knowledge of protein folding, the methods would seem sound. The results are interesting and give insight into the folding of homodimers.
My main criticisms are these:
(1) The heat map presentation is not clear. I can't judge its merit, as I do not understand the graphs. Their meaning is not described, the axes are not labeled. This needs to be fixed. Perhaps these are standard in the computational field, but to reach a wider audience, some background is needed.
(2) Authors should mention how these 8 homodimers were chosen, beyond being 4 two-state and 4 three-state predictions. Were they selected from others to demonstrate the distribution in Table 1, or was this distribution random/unexpected?
(3) It would clarify if paragraph (lines 267-281) numbered the three mechanisms. Also, the three mechanisms can additionally be described as: fold-then-associate, associate-then-fold, one molecule fold-then associate-then other molecule fold. Correspondence between this paragraph and the Figure 2 could also be made more clear. Just a matter of orienting the reader here.
(4) In the discussion, I think everyone already knows that the strict two-state or three-state models are a rough approximation. I think it would be better to acknowledge that this is the generally held view, and that this work corroborates that view and adds specifici detail. I don't think it is stunning, that is an overstatement.
(5) Nowhere is it acknowledged that this is theoretical only. Whether these represent what actually happens with the molecules should be discussed, and experiments should be suggested to corroborate (I am not suggesting doing the experiments for this paper). So how to experimentally detect the folding mechanisms, their weights, their dependence on concentration, etc. And experimentally detect the heat map information, although I don't know what that information is right now.
But overall, an interesting paper that provides insights.
Round 2
Reviewer 1 Report
Comments and Suggestions for Authors
Dear Authors,
I think we are done -- paper is ready to be published, as much as I can judge.
The only non-obligatory comment, never met the word foldedness. Better say folding probability. Up to you, I dont insist and dont need to se the paper again.
Congrats and best wishes...
Referee
Author Response
"I think we are done -- paper is ready to be published, as much as I can judge.
The only non-obligatory comment, never met the word foldedness. Better say folding probability. Up to you, I dont insist and dont need to se the paper again.
Congrats and best wishes..."
Thank you for the suggestion. We think that "foldedness probability" is more accurate as we are talking about the probability of being folded, while "folding probability" suggests the probability of entering the folding process. Thus, we have kept the "foldedness probability" phrase. There are many papers using the word "foldedness" with regards to proteins. Thank you for reviewing our manuscript.
Reviewer 2 Report
Comments and Suggestions for Authors
Thanks, the changes are fine, except for the heat maps. I appreciate the addition of formulae and such, that can go in the methods section, or perhaps it is not needed at all. I rather think it will only further obscure the meaning for the non-expert reader if left where it is.
Instead, I was looking for a concise, easy to grasp, description that lets us know the meaning of what is being shown. The methodology for generating the graph is not the point.
Author Response
Thanks, the changes are fine, except for the heat maps. I appreciate the addition of formulae and such, that can go in the methods section, or perhaps it is not needed at all. I rather think it will only further obscure the meaning for the non-expert reader if left where it is.
Instead, I was looking for a concise, easy to grasp, description that lets us know the meaning of what is being shown. The methodology for generating the graph is not the point.
Thank you for the suggestion. We have added the following text to section 3.3 (lines 345-351):
"The heatmaps in Figure 3 visualize these fluxes. The heatmaps represent the distribution of binding events over the folding degrees of the two chains; the locations of the red areas on a heatmap show us the folding degrees of the chains when most binding events occur. In the heatmap, rigid docking appears in the bottom right corner, conformational selection at the bottom and the right edges of the map, and induced folding appears away from these areas. Thus, the heatmap provides us with a finer-grained picture of the dimer formation process than the simple classification into three mechanisms."
We hope this is an accessible explanation for all readers. We have still left the formula in the text for an unambiguous definition. Thank you again for reviewing our manuscript.